# *PXO_RS20535*, Encoding a Novel Response Regulator, Is Required for Chemotactic Motility, Biofilm Formation, and Tolerance to Oxidative Stress in *Xanthomonas oryzae* pv. *oryzae*

**DOI:** 10.3390/pathogens9110956

**Published:** 2020-11-17

**Authors:** Abdulwahab Antar, Mi-Ae Lee, Youngchul Yoo, Man-Ho Cho, Sang-Won Lee

**Affiliations:** 1Department of Genetic Engineering and Graduate School of Biotechnology, Kyung Hee University, Yongin 17104, Korea; abulwahhab@khu.ac.kr (A.A.); malee0504@khu.ac.kr (M.-A.L.); yooyc@khu.ac.kr (Y.Y.); manhocho@khu.ac.kr (M.-H.C.); 2Crop Biotech Institute, Kyung Hee University, Yongin 17104, Korea

**Keywords:** response regulator, *Xanthomonas oryzae* pv. *oryzae* (*Xoo*), motility, biofilm formation, oxidative stress, type VI secretion system (T6SS)

## Abstract

*Xanthomonas oryzae* pv. *oryzae* (*Xoo*), a causal agent of bacterial leaf blight of rice, possesses two-component regulatory systems (TCSs) as an intracellular signaling pathway. In this study, we observed changes in virulence, biofilm formation, motility, chemotaxis, and tolerance against oxidative stress of a knockout mutant strain for the *PXO_RS20535* gene, encoding an orphan response regulator (RR). The mutant strain lost virulence, produced significantly less biofilm, and showed remarkably reduced motility in swimming, swarming, and twitching. Furthermore, the mutant strain lost glucose-guided movement and showed clear diminution of growth and survival in the presence of H_2_O_2_. These results indicate that the RR protein encoded in the *PXO_RS20535* gene (or a TCS mediated by the protein) is closely involved in regulation of biofilm formation, all types of motility, chemotaxis, and tolerance against reactive oxygen species (ROS) in *Xoo*. Moreover we found that the expression of most genes required for a type six secretion system (T6SS) was decreased in the mutant, suggesting that lack of the RR gene most likely leads to defect of T6SS in *Xoo*.

## 1. Introduction 

*Xanthomonas oryzae* pv. *oryzae* (*Xoo*) is a Gram-negative rod-shaped bacterium and one of the most common pathogens of rice [1]. *Xoo* is a vascular pathogen, invading rice plants through natural openings on leaves, hydathodes, or wounds [2]. *Xoo* multiplies in xylem and impedes water flow, resulting in drying of leaves. A disease caused by *Xoo* is bacterial leaf blight (BLB), which causes significant loss of potential rice yield worldwide [3]. This indicates the need for in-depth study of virulence mechanisms of the bacteria and efforts over the past decades have functionally characterized many of the virulence factors required for bacterial disease. The factors include diffusible signal factors (DSFs), extracellular polysaccharides (EPSs), lipopolysaccharides (LPSs), adhesins, motility factors, biofilm formation species, type 2 secretion system (T2SS), type 3 secretion system (T3SS), type 6 secretion system (T6SS), and extracellular enzymes [4,5,6,7,8,9,10,11,12].

Bacteria rapidly sense and respond to diverse environmental conditions to ensure their survival. This immediate biological response is also important for growth and propagation in host conditions of pathogenic bacteria such as *Xoo*. The representative regulatory system for such biological processes is the two-component regulatory system (TCS). The TCS controls a broad spectrum of biological processes including pathogenesis in bacteria. A typical TCS functions through two-proteins; one is a histidine protein kinase (HK) sensor containing a cytoplasmic kinase core and a periplasmic N-terminal sensing domain, and the other is a response regulator (RR) with receiver and effector domains. Signal recognition by HK causes auto-phosphorylation at a histidine residue in the sensor protein, and a high-energy phosphate group subsequently is transferred to the aspartate residue in the cognate RR protein. The activated RR regulates its downstream gene expression as a transcriptional regulator or directly interacts with the target proteins [13,14]. The TCS genes account for about 3% of the total nucleotide sequence in *Xoo* genome, few of which including RpfG/RpfC, HrpG, RaxR/RaxH, PhoP/PhoQ, ColR/ColS, and DetR have been functionally characterized in relation to the pathogenicity of *Xoo* [15,16,17,18,19]. A TCS mediated by RpfC/RpfG was first identified in *Xanthomonas* spp. and is well-conserved in all xanthomonads [4,15]. The DSF signals mediated by RpfC/RpfG promote EPS production and xylanase activity in *Xoo* [4]. The HrpG, which is stimulated by unknown signals, is the critical regulator of *hrp* (hypersensitive responses and pathogenicity) genes. HrpG-HrpX (transcriptional activator) controls genomewide regulons, including *hrps*, T3SS genes, T3SS effectors, T2SS genes, and putative virulence genes [16,20]. *Xoo* requires a TCS mediated by RaxR/RaxH to regulate the expression of *rax* genes for AvrXA21 activity and to sense population cell density. AvrXa21 is an avirulence protein produced by *Xoo* and is the ligand of XA21, the most well-known receptor in plant immune systems [21]. Another TCS, PhoP/PhoQ, also is required for AvrXA21 activity and *hrp* gene expression and is negatively regulated by RaxR [22]. The ColR/ColS system is required for *Xoo* virulence in rice plant to promote growth in iron-limited conditions and expression of the T3SS component genes [18]. Recently, we reported a RR protein, DetR that is essential for *Xoo* virulence through the regulation of EPS synthesis, reactive oxygen species (ROS) detoxification, and iron homeostasis [19]. The gene *PXO_RS05125* encoding DetR protein was identified from screening with 62 *Xoo* RR gene mutant strains [19,23]. 

In the present study, we focused on *PXO_RS20535* (tentatively named *rr35* in this study) gene, identified from the screening using *Xoo* RR gene mutant strains [23]. We first carried out RNA-sequencing analysis with a knockout mutant strain (RR35) and the wild-type strain (PXO99A). Based on data set from the RNA-sequencing analysis, we investigated phenotypic changes and genes expression of the mutant strain to elucidate regulatory function of RR35 protein (or a TCS mediated by RR35) in *Xoo* pathogenicity.

## 2. Materials and Methods 

### 2.1. Biological Materials and Growth Conditions

All bacterial strains and primers used in this study are listed in Appendix A. *Xoo* strains were grown at 28 °C in peptone sucrose broth (PSB) as a rich medium and in XOM2 as a host mimic medium [0.18% d-xylose, 670 μM d,l-methionine, 10 mM sodium l(+)-glutamate, 14.7 mM KH_2_PO_4_, 40 μM MnSO_4_, 240 μM Fe(III)-EDTA (ethylenediaminetetraacetic acid), and 5 mM MgCl_2_] [24].These media often contain suitable antibiotics such as 15 μg/mL of cephalexin for selection of *Xanthomonas*, 50 μg/mL of kanamycin for selection of the RR35 mutant strain, and 10 μg/mL of gentamycin for selection of a complementary strain of RR35 (cRR35). 

Dong-jin (a japonica rice cultivar compatible with PXO99A) seeds were germinated in water at room temperature for 3 to 4 days, transferred into pots (12 cm in diameter and 11 cm in height) filled with soil supplied from rice paddy fields, and then grown in a greenhouse for five weeks (28 °C day/22 °C night). Five-week-old plants (generally the six-leaf stage) were transferred to a growth chamber at least two days prior to inoculation. The chamber conditions were 14/10 h of light/dark at 28/25 °C and 80/90% relative humidity.

### 2.2. Virulence Assays

We inoculated the PXO99A, RR35, and cRR35 strains using the leaf clipping method as previously described [23,25]. *Xoo* strains were cultured for 3 days at 28 °C on peptone sucrose agar (PSA) medium containing suitable antibiotics for each strain and were suspended in water to achieve a population of 10^7^ CFU (colony forming units)/mL. Each bacterial suspension was inoculated in Dong-jin rice leaves, and lesion lengths were measured after 14 days. Virulence assays with RR35 and cRR35 have been repeated dozens of times over the past two years, and the results have been consistent.

### 2.3. RNA Extraction

PXO99A and RR35 strains were grown in XOM2 broth medium to the mid-exponential phase (OD_600_ = 0.5) in a shaking incubator at 28 °C. Total RNA was extracted using RNeasy Plus Mini Kit (Qiagen, Valencia, CA, USA) according to the manufacturer’s instructions, and the RNA pellet was suspended in 50 µL of DEPC (diethyl pyrocarbonate)-containing distilled water. The concentration and quality of each RNA sample were assessed using the Nanodrop Spectrometer ND-2000 (Thermo Scientific, Waltham, MA, USA).

### 2.4. RNA-Sequencing Analysis

Extracted RNA from each strain was qualified using TapeStation RNA screen tape (Agilent). Only high-quality RNA preparations, with RNA integrity number greater than 7.0, were used for RNA library construction. A library was prepared with 1 ug of total RNA of each sample by Illumina TruSeq RNA Sample Prep kit (Illumina, Inc., San Diego, CA, USA). The XOM2-derived libraries were sequenced using the NovaSeq platform (Illumina) by Macrogen Inc. (Seoul, South Korea). We preprocessed the raw reads from the sequencer to remove low quality and adapter sequence before analysis and aligned the processed reads to the *Xoo* PXO99A (GCF_000019585.2) using Bowtie 1.1.2. The reference genome sequence of *Xoo* PXO99A (GCF_000019585.2) and annotation data were downloaded from the NCBI (National Center for Biotechnology Information; https://www.ncbi.nlm.nih.gov/assembly/GCF_000019585.2/). Transcript assembly and abundance estimation using HTSeq. After alignment, HTSeq v0.10.0 was used to assemble aligned reads into transcripts and to estimate their abundance. It provides the normalized estimates as RPKM (reads per kilobase of transcript per million mapped reads) values of transcript and gene expressed in each sample. These values are used for the comparison of differentially expressed genes between PXO99A and RR35 strains.

We performed statistical analysis to identify differentially expressed genes using estimates of abundance for each gene in the samples. Genes with one more than zero RPKM value were excluded. To facilitate log2 (gene expression change comparing RR35 vs. PXO99A) transformation, a unit of 1 was added to the RPKM value of each filtered gene. Filtered data were log2-transformed and subjected to quantile normalization. Statistical significance of the differential expression data was determined using an independent *t*-test and fold change analysis, in which the null hypothesis was no difference among groups. False discovery rate was controlled by adjusting the *p*-value using the Benjamini–Hochberg algorithm. For differentially expressed genes, hierarchical clustering analysis was performed using complete linkage and Euclidean distances as measures of similarity. Gene-enrichment and functional annotation analyses and pathway analysis for significant gene lists were performed based on DAVID (database for annotation, visualization, and integrated discovery; https://david.ncifcrf.gov/) and the KEGG (Kyoto Encyclopedia of Genes and Genomes) pathway (http://www.genome.jp/kegg/pathway.html).

### 2.5. Biofilm Formation Assay 

Quantification of biofilm formation of each strain was carried out using borosilicate glass tubes as described previously [26]. *Xoo* strains were incubated in PSA and seeded in XOM2 broth containing suitable antibiotics at 28 °C with agitation. Three milliliters of the bacterial suspension was added to each borosilicate glass tube at a starting OD_600_ (optical density at 600 nm) of 0.05 and incubated without shaking at 28 °C. After four days, planktonic cell density (OD_600_) was measured using an ultraviolet (UV) spectrophotometer (UV1800 Shimadzu, Nakagyoku, Kyoto, Japan), and the planktonic cells were removed. Borosilicate glass tubes were washed three times with distilled water, the attached bacterial cells were stained with 0.1% crystal violet for 30 min, and each tube was washed three times with distilled water. The amount of crystal violet in ethanol was spectrometrically quantified at 550 nm. Biofilm forming activity (OD_550_/_600 nm_) was calculated by normalization to bacterial growth of each tube. This analysis was carried out more than three times with independently prepared samples. All results were consistent. 

### 2.6. Swimming, Swarming, and Twitching Motility Assays

For the swimming and swarming motility assays, bacterial cells of each strain were cultured in XOM2 broth medium in a shaking incubator at 28 °C for 2 days to OD_600_ of 0.2. Bacterial cells of each strain were dropped onto XOM2 plates containing 0.3% and 0.5% agar for swimming and swarming motility assays, respectively. The swim and swarm plates were placed at 28 °C for 7 days, and the diameters of halo zones were measured to observe the motile ability of strains. For monitoring of the growth of controls, the strains were dropped onto 1.5% XOM2 agar plates and incubated at the same conditions (Appendix A). The twitching motility assay was done by pouring XOM2 medium (1% agar) into borosilicate glass tubes. Each strain was incubated in PSA medium at 28 °C for 2 days, harvested and washed several times with XOM2 broth, and suspended with XOM2 broth. For inoculation, a needle was dipped into the suspension and stabbed onto solid XOM2 in a borosilicate glass tube. Twitching motility of each strain was monitored for 7 days at 28 °C.

### 2.7. Chemotaxis Assay

Chemotaxis of the RR35 strain was investigated on the surface of semisolid soft agar medium 0.3% without a carbon source, as previously reported with slight modification [27]. Briefly, *Xoo* strains were cultured in XOM2 broth medium in a shaking incubator at 28 °C to OD_600_ = 3.5 × 10^8^ CFU/mL, transferred to XOM2 fresh medium, and adjusted to OD_600_ = 0.2. Next, sterilized filter discs (5 mm diameter) were infiltrated with 15% (wt/vol) glucose or water under vacuum and placed in the center of a soft agar plate (35 × 10 mm, SPL life science, South Korea). Two microliters of each cell strain were dropped on either side, 1 cm away from the paper disc. The plates were placed at 28 °C, and movement of bacteria toward the disc was monitored for 3 to 5 days. 

### 2.8. H_2_O_2_ Tolerance Assay

To test the effect of hydrogen peroxide (H_2_O_2_) on growth of each strain, we used a previously reported method [28] with slight modification. Each strain was cultured in XOM2 broth medium at 28 °C with shaking until the population reached 3.5 × 10^8^ CFU/mL. Then, 200 μL of each cell culture was mixed with 20 mL PSA medium (1.0% agar) containing appropriate antibiotics and poured into a Petri-dish (90 × 15 mm, SPL life science, Seoul, South Korea). After solidification, a 6-mm-diameter sterilized Whatman paper disc was placed on the center of the plate, and then 5 μL of H_2_O_2_ (1 mM and 10 mM) was dropped onto the paper disc. Halo zone formation by H_2_O_2_ on the PSA plate was monitored, and the diameter was measured after 3 days of incubation at 28 °C. 

Survivability of each strain against ROS was tested as described previously [19]. Each strain was incubated in XOM2 broth medium at 28 °C with shaking to 3.5 × 10^8^ CFU/mL. These cells were transferred to fresh XOM2 containing H_2_O_2_ (0, 1, or 10 mM) at a density of OD_600_ = 0.2 and then shaken at 200 rpm and 28 °C for 6 h. The cells were spotted after serial dilution onto PSA plates containing appropriate antibiotics to count cell number.

### 2.9. Quantitative Real-Time Reverse-Transcription Polymerase Chain Reaction (qRT-PCR)

To analyze the expression of related genes of each virulence factor, quantitative real-time reverse-transcription polymerase chain reaction (qRT-PCR) was carried out with specific primer sets for each gene, as shown in Appendix A. Total RNA from each strain was extracted and used for cDNA synthesis using a Prime Script RT reagent kit (TaKaRa, Otsu, Shiga, Japan). To estimate real-time qRT-PCR, each mixture containing TB Green Premix Ex Taq (TaKaRa, Otsu, Shiga, Japan) was analyzed using a Rotor Gene Q (Qiagen, Venlo, Netherlands). The 16S rRNA was used as a reference gene in each experiment. Relative expression of genes was quantified using the 2^−∆∆CT^ method [29].

### 2.10. Arithmatic Measurement of Experimental Data and Statistical Analysis

All data from experiments in this study express the difference from comparative analysis between strains of a wild-type (PXO99A) and a knock-out mutant (RR35) that does not have a gene (*rr35*) in the PXO99A. Bacterial strains (PXO99A and RR35) were prepared for three replicates for each experiment under the same conditions and average ± standard deviation outcome of the three replicates of each strain are expressed as a bar in each graph. All the experiments in this study were repeated more than three times with consistent results and all figures in this report show a representative result of the repeats for each experiment. Bars in Figure 6, showing qRT-PCR results of genes, reveal the expression level of each gene in the mutant strain relative to those in the PXO99A strain. Statistical significance was analyzed by one-way ANOVA and Student’s *t*-test using GraphPad Prism 7.0 (GraphPad Software, San Diego, CA, USA).

## 3. Results

### 3.1. The rr35 (PXO_RS20535) Gene Is Required for Virulence of Xoo

From our mutant screening, we selected six *rr* genes, associated with *Xoo* virulence [23]. The *rr35* gene used in this study is one of the six. Consistent with our previous report [23], the lesion length on leaves of Dong-jin rice inoculated with the *PXO_RS20535* gene knockout strain was significantly decreased (3.51 ± 1.30 cm) compared to that of wild-type strain PXO99A (9.22 ± 2.54 cm) in several repeated inoculation tests (Figure 1). A complementary strain, RR35 mutant strain carrying pBBR1-MCS5 inserted by 6 X *His*-*rr35* (cRR35), showed a longer lesion length (9.24 ± 2.64 cm) than the RR35 strain, similar to that of PXO99A inoculation (Figure 1). This result indicates that the cRR35 strain restored virulence, and the reduced virulence observed in the RR35 mutant strain was due to knockout of the *PXO_RS20535* gene, not to unexpected mutation of adjacent genes during homologous recombination in mutagenesis. Furthermore, we established growth curves of these three strains (PXO99A, RR35, and cRR35) in plant-mimicking XOM2 medium to test if the reduced virulence of RR35 strain was due to decreased ability in growth metabolism (Appendix A). The growth rate of RR35 was not different from that of PXO99A or cRR35, suggesting that the *rr35* gene is required for virulence but not for essential *Xoo* biological functions such as growth.

### 3.2. The rr35 (PXO_RS20535) Gene Is Highly Conserved in All Xanthomonas spp.

The *rr35* gene encodes an orphan RR protein (128 amino acids) belonging to the CheY family. In silico analysis with an amino acid sequence deduced from the *rr35* gene revealed that it encodes a well-conserved protein in most *Xanthomonas* spp., annotated as twitching motility RR PilH in the public database (https://www.ncbi.nlm.nih.gov/). However, this protein is distinct from the well-studied PilH protein of *Pseudomonas aeruginosa* (*P. aeruginosa*) and *Xanthomonas campestris* pv. *campestris* (*Xcc*). The RR35 protein showed only 55.46% and 52.21% identities with amino acid sequences of the PilH protein of *P. aeruginosa* and *Xcc*, respectively (data not shown). In the genome, required genes for type IV pili of *P. aeruginosa* and *Xcc* are clustered in an operon, *pilG-pilH-pilI-pilJ-pilK*. However, *rr35* in the genome is independent of other *pil* genes. These findings indicate that the RR35 protein either regulates *Xoo* pilus independently or has a different function from PilH of *P. aeruginosa* and *Xcc*.

### 3.3. RNA-Sequencing Analysis Showed that RR35 Regulates Genes Involved in Biofilm Formation, Motility, and Tolerance against Oxidative Stress in Xoo

To find DEGs (differential expression genes) between PXO99A and RR35 strains, Illumina RNA-sequencing analysis was carried out. Totals of 30,538,082 and 32,836,468 sequence reads were respectively generated from PXO99A and RR35, and 65 DEGs including hypothetical proteins that were finally selected (Table 1). Forty of the 65 genes showed decreased expression in the RR35 mutant strain with an RPKM (reads per kilobase per million) value of about 2, suggesting that expression of these genes is directly or indirectly activated by RR35 protein (or a TCS mediated by RR35). Some of the 40 genes are related with cellular response to H_2_O_2_, motility, biofilm formation, or synthesis of virulence factors (*katE*, *PXO_RS17325*; *catB*, *PXO_RS22725*; fimbrial protein, *PXO_RS16000*; DNA-3-methyladenine glycosylase I, *PXO_RS17855*; acetyl-CoA desaturase, *PXO_RS01535*) in other *Xanthomonas* spp. and/or other pathogenic bacteria, such as *Listeria monocytogenes* [20] and *P. aeruginosa* [30]. We validated RNA-sequencing data with 29 genes that were randomly picked and known to be involved in virulence of pathogenic bacteria using RT-PCR and/or real-time RT-PCR (Appendix A). Expression pattern of 21 in the 29 genes were agreed with RNA-sequencing data. Although fidelity of the transcriptome analysis was not very high and DEGs did not provide many candidates, the results suggest that RR encoded in *PXO_RS20535* likely controls the expression of genes crucial for biofilm formation, motility, and tolerance against oxidative stress in the host.

### 3.4. RR35 Mutant Strain Forms Much Less Biofilm than the Wild-Type Strain PXO99A

To test if RR35 modulates biofilm formation in *Xoo*, we measured the amount of RR35 mutant strain cells attached to the surface of borosilicate glass and compared it with that of PXO99A. After 4 days of culture in borosilicate glass tubes, the amount of attached RR35 strains was smaller than those of PXO99A and cRR35 strains (Figure 2A). On quantification of the pellicle dissolved in crystal violet after four days, the RR35 strain showed less than half the ability for biofilm formation (OD_550/600_ = 0.57 ± 0.08), normalized by dividing by planktonic cell mass, of those of wild-type PXO99A (1.31 ± 0.08) and complementary strain cRR35 (1.57 ± 0.09) (Figure 2B). These results suggest that the RR35 protein is associated with the regulation of biofilm formation of *Xoo.*


### 3.5. All Modes of Bacterial Motility Are Reduced in the RR35 Mutant Strain

The *rr35* gene is annotated to encode a twitching motility RR protein, and the RR35 mutant strain has decreased expression of a fimbrial protein (Table 1), showing high homology to type IV fimbrial biogenesis protein FimT of *Xanthomonas* spp. Type IV fimbriae are associated with twitching motility, which is involved in bacterial translocation across solid surfaces [31]. Motility of swimming, swarming, and twitching of the RR35 mutant strain was examined and compared with those of PXO99A and cRR35 strains. In conclusion, lack of *rr35* in *Xoo* caused decrease in both flagellum-dependent and pili-dependent motility (Figure 3). PXO99A and cRR35 strains spread from the inoculation origin faster than the RR35 strain on the surface of a 0.3% agar plate (inset pictures of Figure 3A). Diameter of swimming halo zones of the RR35 strain was much smaller (4.5 ± 0.71 mm) than those of PXO99A and cRR35 strains (11.50 ± 0.71 mm and 14.50 ± 0.71 mm, respectively) (Figure 3A). The RR35 strain also showed clear decrease in twitching motility (Figure 3B). Stabbed PXO99A and cRR35 widely moved through all area of the tubes, while RR35 strains remained at the stabbed origin of the tube (Figure 3B). This mutant strain also showed decreased multicellular movement on a 0.5% agar plate (Figure 3C,D). The PXO99A and cRR35 strains showed a halo zone with 14 ± 0.81 mm and 16.5 ± 1.29 mm diameters, respectively, while that of the RR35 mutant strain was 9.5 ± 0.57 mm (Figure 3D). These results indicate that the RR35 protein regulates *Xoo* swimming, swarming, and twitching motility, all of which are regarded as virulence factions of pathogenic bacteria.

### 3.6. Chemotaxis-Guided Movement Is Lacking in the RR35 Mutant Strain

The RR35 protein belongs to the CheY family that is known to be involved in switching of flagella motor protein by chemotactic signals recognized by a chemoreceptor, CheA [32,33]. Although *rr35* is an orphan RR gene and could not associated with a cognate HK in this study, deficiency of the *rr35* gene possibly causes changes of chemotaxis-guided movement in *Xoo*. Therefore, we tested chemotaxis ability of the RR35 strain using a disc assay. Fifteen percent (wt/vol) glucose was used as a chemoattractant, and distilled water was used as a negative control (Figure 4). No strains showed any movement in a specific direction on the plates having a water-soaked paper disc, but PXO99A and cRR35 strains showed clear movement toward the 15% glucose-saturated disc (Figure 4). However, the RR35 mutant strain did not show movement on the plate with a 15% glucose-saturated disc. This result implies that lack of the *rr35* gene in *Xoo* resulted in loss of the chemotactic response.

### 3.7. The RR35 Strain Is Highly Sensitive to H_2_O_2_

The *katE* and *catB* are genes encode ROS detoxifying enzymes [28], and their expression was reduced in the RR35 mutant strain according to RAN-sequencing analysis. This indicates that the mutant strain is more sensitive to oxidative stress. To examine this, PXO99A, RR35, and cRR35 strains were exposed to 0, 1, and 10 mM of H_2_O_2_. As shown in Figure 5A,B, each strain produced a larger inhibitory zone in the higher concentration (10 mM) of H_2_O_2_. At each H_2_O_2_ concentration, RR35 showed the largest halo zone. The diameter of the inhibitory zone of PXO99A was 16 ± 1 mm in 1 mM H_2_O_2_ and 38 ± 0.6 mm in 10 mM H_2_O_2_. Similarly, halo zone diameter of the cRR35 strain was 15 ± 0.6 mm and 36 ± 0.6 mm, respectively. However, diameters of the inhibitory zone for the RR35 strain were almost twice those of the others (25 ± 0.6 mm in 1 mM and 56 ± 1.1 mm in 10 mM H_2_O_2_). This result illustrates that growth of the mutant strain RR35 is much more sensitive to H_2_O_2_ than is that of the wild-type strain PXO99A. Survivability of each strain in the presence of H_2_O_2_ was tested (Figure 5C,D). For this experiment, a high population density of each strain was suspended in H_2_O_2_-containing XOM2 broth medium, and the population of each strain was monitored every hour for 6 h. Over time, populations of all strains decreased in the presence of H_2_O_2_ compared to that in XOM2 medium without H_2_O_2_. RR35 was most severely decreased (Figure 5C,D), with a population of 1.7 × 10^8^ CFU and 8 × 10^7^ CFU/mL in 1 mM and 10 mM H_2_O_2_ at 6 h after inoculation, respectively, while those of PXO99A and cRR35 were 3 × 10^8^ and 3.5 × 10^8^ CFU/mL in 1 mM and 1.7 × 10^8^ and 2.2 × 10^8^ CFU/mL in 10 mM H_2_O_2_-containing medium after 6 h. This result suggests that *rr35* confers tolerance to growth and survival under conditions of H_2_O_2_.

### 3.8. Expression of Genes Involved in Biofilm Formation, Motility, and H_2_O_2_ Tolerance Was Reduced in the RR35 Mutant Strain

In this study, the RR35 strain showed impaired biofilm formation as well as defective in motility, chemotaxis, and resistance to H_2_O_2_. We tested the expression of genes known to be involved in the changed phenotypes, even if this was not shown in RNA-sequencing data, with qRT-PCR using the specific primers listed in Appendix A. 

Biofilm formation is an essential virulence factor in which pathogenic bacteria defend themselves against various environmental stresses [34]. Flagella-dependent motility is necessary during initial biofilm formation to identify and interact with an appropriate surface [35]. To understand the reduced motility and biofilm formation ability of RR35, we tested the transcriptional expression of *fleQ* (*PXO_RS11980*), which is the master regulator in the gene expression associated with biosynthesis of a flagellum with alternative sigma factor RpoN2 [27,36]. As shown in Figure 6, the transcriptional level of *fleQ* (*PXO_RS11980*) decreased in the RR35 strain, suggesting that the expression of the master regulator of flagellar genes is influenced by RR35. Moreover, we examined the expression of genes required for flagellar production, including *fliA* (*PXO_RS12145*)*,* RNA polymerase sigma factor of motility; *fliD* (*PXO_RS12875*), flagellar cap protein; *fliC* (*PXO_RS12870*), flagellin; and *fliTX* (*PXO_RS12885*), which is necessary for flagellar biogenesis and motility. These genes are FleQ-regulated and were decreased similarly to *fleQ* in RR35 (Figure 6). Regarding biofilm formation, few genes were identified as direct contributors. Recently, the major phosphodiesterase EdpX1 was shown to be closely related to biofilm formation [37]. In the RR35 strain, the *epdX1* gene (*PXO_RS01540*) was reduced compared with that of PXO99A (Figure 6). We concluded that RR35 protein (or a TCS mediated by RR35) positively controls gene expression for motility and biofilm formation in *Xoo*. 

ROS including hydrogen peroxide recently has been acknowledged as a molecule for plant defense responses against pathogens [38,39]. As a phytopathogenic bacterium, *Xoo* should immediately overcome oxidative stress after infection by activation of detoxifying mechanisms. In *Xoo*, OxyR is a critical regulator of expression of the catalase genes *katE* (*PXO_RS17325*), *catB* (*PXO_RS22725*), and *srpA* (*PXO_RS22555*) [28,40] In our test, expression of *oxyR* (*PXO_RS05465*) and *srpA* (*PXO_RS22555*) was slightly increased in the RR35 mutant strain, while *katE* (*PXO_RS17325*) and *catB* (*PXO_RS22725*) genes showed significant reduction in expression (Figure 6). These results suggest that the RR35 protein positively regulates those catalase genes, contributing to H_2_O_2_ neutralization, but that such regulation is not involved in the OxyR pathway.

Gene expression for T6SS was examined because a gene encoding the type VI secretion system tip protein VgrG was expressed significantly higher in the RR35 mutant strain than in PXO99A (Table 1). This suggests that RR35-mediated TCS can positively control T6SS. Therefore, we chose many genes known to be required for T6SS and carried out qRT-PCR. All genes (*tssE*, *PXO_RS07340*; *tssD*, *PXO_RS07330*; *tssL*, *PXO_RS07400*; *vgrG*, *PXO_RS07300* & *PXO_RS17555*; *tssH*, *PXO_RS07355*) showed reduced expression in the RR35 mutant strain compared to that in PXO99A (Figure 6). This suggests that the RR35 protein positively regulates T6SS, and deficiency of T6SS caused by knocking-out of *rr35* might be one of reasons for the reduced virulence in the RR35 strain.

## 4. Discussion

The TCSs regulate most biological processes in bacteria, including the virulence of pathogenic bacteria. However, little is known about TCSs, which control the pathogenicity of *Xoo* that invades rice plants and causes bacterial leaf blight disease. We generated 55 knockout mutant strains targeting 62 genes, annotated as RR genes of a *Xoo* strain PXO99A in a public database (Prokaryotic Two-Component System, http://www.p2cs.org/) and have found 6 new *RR* genes required for *Xoo* virulence [23]. One of the six is *detR*, regulating expression of enzymes to provide tolerance to *Xoo* against oxidative stresses [19]. In this study, we focus on another one of the six, tentatively named *rr35.* This gene is located at *PXO_RS20535* without a cognate HK and is well-conserved in *Xanthomonas* spp. as the twitching motility RR gene *pilH*. This protein is suggested to function as an RR for twitching-dependent motility in *Xanthomonas*, but has not yet been characterized. Twitching motility system is well-studied in *P. aeruginosa* [41,42]. In the genome of *P. aeruginosa*, *pilH* is located in the *pil* cluster consisting of the *pilG*-*pilH*-*pilI*-*pilJ*-*pilK* operon. The PilH protein participates in control of a type IV pilus function with another RR protein PilG through a signal from the HK ChpA [43]. It has been known that PilH and PilG antagonistically regulate flagellum-dependent and pili-dependent motility in *Xcc* [44]. This means that PilH positively controls swarming motility, while negatively regulates swimming motility in *Xcc*. In silico analysis reveals that the *rr35* gene is not a member of the *pil* cluster, even though it is annotated with the same name. *rr35* is located independently in the *Xoo* genome, not in the *pil* operon. In addition, the RR35 protein did not have significant amino acid sequence identity with PilH proteins of *P. aeruginosa* and *Xcc.* Instead, *PXO_RS14970* (RR22 in our list [23]) corresponds as a homologous gene with *pilH* of *P. aeruginosa* and *Xcc*, with 48.7% and 100% identities, respectively. Moreover, the PilG-homologous gene (*PXO_RS14975*) was located downstream of the *PXO_RS14970* gene in *Xoo*. This suggests that the *rr35* gene is somewhat homologous in sequence with PilH, but both are distinct. Furthermore, RR35 positively regulates both swimming and swarming motilities in *Xoo* (Figure 3), unlikely to *Xcc* PilH. Lack of the *rr35* gene caused clear reduction in all modes motility including swimming, swarming, and twitching. This is a reason why *PXO_RS20535* is not named *pilH* in this study. 

Such bacterial movement is not random and is directed by certain beneficial or toxic chemicals [45]. This movement in response to a chemical stimulus is called chemotaxis. Typically, HK chemotaxis receptor protein CheA receives signals from stimuli and transfers them to two RR proteins, CheY and CheB. Among them, CheY controls flagellar motor switching [46]. The RR35 mutant strain showed loss of chemotaxis-guided movement (Figure 4). This could be a fatal defect of the RR35 strain that would be in competition with host cells. After infection, *Xoo* must constantly sense environmental conditions for nutrients and move rapidly to a better location. However, our results illustrated that the RR35 strain lost the ability to move toward glucose. 

Our RNA-sequencing data suggest that RR35-mediated signaling regulates biofilm formation, H_2_O_2_ tolerance, and T6SS biogenesis (Table 1 and Appendix A). Biofilms in microorganisms are required for metabolic cooperation. Defense improvement, colonization, increase of nutrient availability, multiplication, communication by signaling materials, adhesion on biotic/abiotic surfaces, and invasion of a host are representative functions of biofilm [47,48]. Biofilm formation is an essential virulence factor of pathogenic bacteria [34]. Despite this importance, few such genes have been identified in *Xoo*. In *Xoo* spp. RpfC/RpfG, a TCS consisting of the sensor kinase RpfC and the RR RpfG, leads to change in production of EPS, extracellular enzymes, and biofilm formation [49,50]. In recent report [37], EAL domain protein EdpX1 was shown to be a major phosphodiesterase to regulate virulence phenotypes including EPS production and biofilm formation depending on the c-di-GMP signaling pathway. In our test, attachment of RR35 mutant cells to the wall of the borosilicate glass tube was reduced compared to that of PXO99A and cRR35 strains (Figure 2). Moreover, the gene (*PXO_RS01540*) encoding the *Xoo* EpdX1 was decreased in the RR35 strain (Figure 6). These results reveal that the RR35 protein positively modulates EdpX1 expression and biofilm formation, although we could not clarify the detailed mechanism.

Another key factor associated with virulence of pathogenic bacteria is tolerance against oxidative stresses. ROS, including superoxide (O_2_^−^) and H_2_O_2_, had been thought as toxic byproducts in plants but have been acknowledged as vital molecules in defense against pathogenic bacteria in plants [38,51]. This means that pathogenic bacteria invading host plants encounter a high concentration of molecules such as H_2_O_2_ and should survive under these conditions by triggering a detoxifying mechanism [28,52]. *Xoo* possesses protective proteins detoxifying ROS, such as CatB, KatE, and SrpA [53,54]. In addition, these proteins are directly regulated by the H_2_O_2_-sensing transcriptional regulator OxyR [2]. The gene expression test in this study showed *katE* (*PXO_RS17325*) and *catB* (*PXO_RS22725*) to be significantly decreased in expression compared to that in PXO99A, but *oxyR* (*PXO_RS05465*) and *srpA* (*PXO_RS22555*) were not (Figure 6). *oxyR* (*PXO_RS05465*) and *srpA* (*PXO_RS22555*) were slightly increased, although the changes were not significant statistically. This might be a response to compensate for the decrease of ROS tolerance due to lack of the RR35 protein. This result indicates that RR35 positively regulates expression of main catalases such as CatB and KatE but not that of SrpA. Moreover, regulation of expression of CatB and KatE has no connection with the OxyR regulation pathway. 

Bacterial T6SS is a nanomachine to kill prokaryote and eukaryote cells through transmission of toxic effectors into target cells [55]. Like T3SS and T4SS, T6SS delivers effectors directly to the cytoplasm of target cells [56]. The apparatus of T6SS consists of 13 core proteins (TssA to TssM) and a PAAR (proline-alanine-alanine-arginine)-repeat protein. Among them, TssA, TssE, TssF, TssG, and TssK are component of baseplates; TssJ, TssL, and TssM are membrane complexes; and TssB and TssC are sheath components [57]. TssD is a hemolysin-coregulated protein (Hcp) that acts as a building tube for delivery of effectors from cytoplasm of bacteria to cytoplasm of host cells. The last component is a trimeric VgrG spike on the tip of the Hcp tube, acting as a puncturing system [58]. Most T6SS function in virulence in pathogenic bacteria [59], but connection between T6SS and *Xoo* virulence had been unclear. However, the function of T6SS for *Xoo* virulence was recently reported [11]. The PXO99A genome has two clusters encoding genes required for T6SS. Mutant strains with deletion of the *hcp2* gene in one of two genes clusters (T6SS-2) and a double mutant strain with deletion of both *hcp* genes at clusters T6SS-1 and T6SS-2 in *Xoo* show reduced virulence, but *hcp1*-deletion mutant strain has no change in virulence. Although there is no evidence or possible explanations why only one of these two has an effect on *Xoo* virulence, it is clear that T6SS contributes to virulence of *Xoo*. RNA-sequencing data of RR35 mutant strain showed a VgrG as an up-regulated protein by RR35 protein. This and other genes in the T6SS-1 (VgrG1, *PXO_RS07300*; TssD1, *PXO_RS07330*; TssE1, *PXO_RS07340*; TssH1, *PXO_RS07355*; TssL1, *PXO_RS07400*) and T6SS-2 (VgrG3, *PXO_RS17555*; TssB2, *PXO_RS17625*) clusters were confirmed by RT-PCR to be decreased in the RR35 mutant strain, suggesting that the RR35-mediated TCS most likely regulates T6SS biogenesis.

In this study, we functionally characterized a new RR required for *Xoo* virulence using a knockout mutant strain of the gene *PXO_RS20535*. The mutant strain RR35 showed defects in biofilm formation, all modes of motilities, chemotaxis, and tolerance against H_2_O_2_. These critical defects in the pathogenic bacteria explain the lost virulence of this mutant strain. Together with the gene expression results shown in Figure 6, we conclude that RR35 or a TCS mediated by RR35 (although the full signaling pathway from an HK could not be identified in the present study) is a key regulator of genes encoding virulence factors. Moreover, we report that this RR protein encoded in *PXO_RS20535* is a direct or indirect regulator for T6SS in *Xoo*. As mentioned in our previous reports [19,23], we are elucidating the pathogenic mechanisms of the six RRs that we recently discovered. *PXO_RS20535*, in this study, is the second-characterized *RR* gene after *detR* [19] and research on the remaining four will be completed shortly. Furthermore, to accumulate data on the interrelationships between all RRs in gene expression, we are carrying out large-scale transcriptome analysis, such as RNA-sequencing using RR mutant strains. These further studies will undoubtedly contribute to uncovering the integrated mechanism of TCSs for *Xoo* pathogenesis.

## Figures and Tables

**Figure 1 pathogens-09-00956-f001:**
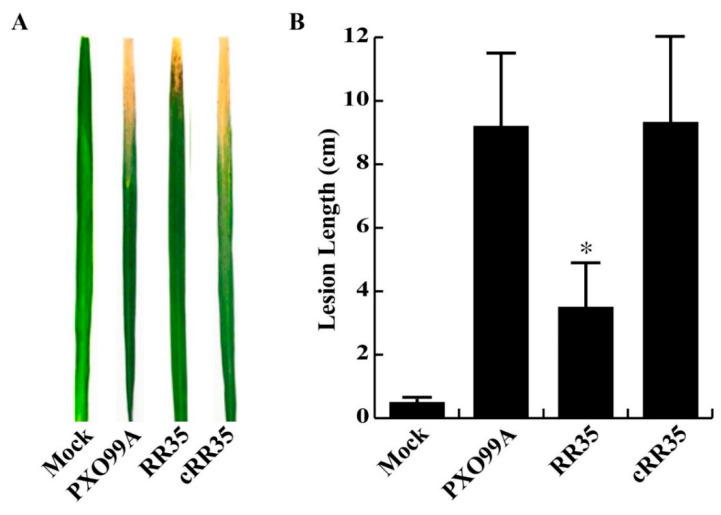
Virulence phenotypes of PXO99A, RR35 and, cRR35 strains. (**A**) Symptoms induced by PXO99A, RR35 and cRR35 on Dong-Jin rice leaves at 14 days post-inoculation (dpi). Rice leaves were infected with 10^7^ colony-forming units (CFU)/mL of bacterial suspension by the scissor clipping method. Bars are mean ± standard deviation (SD) (n = 20). No difference in lesion length was observed on rice leaves inoculated with water (Mock). (**B**) Lesion lengths of Dong-Jin leaves infected with PXO99A, RR35, and cRR35 strains at 14 dpi. * indicates that the lesion length of RR35 was significantly shorter than that of PXO99A and cRR35 by Duncan’s test (*p* < 0.001). This inoculation test was repeated dozens of times with high consistency, and results from one experiment are shown.

**Figure 2 pathogens-09-00956-f002:**
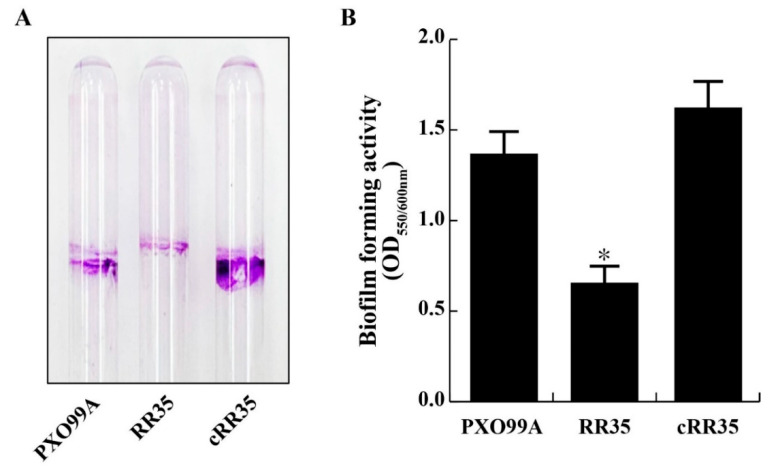
Biofilm formation assays. (**A**) Biofilm formed in borosilicate glass tubes by the PXO99A, RR35, and cRR35 strains. (**B**) Each strain was incubated in borosilicate glass tubes containing XOM2 minimal medium for four days at 28 °C without shaking, and the amount of crystal violet staining was measured using a spectrophotometer (OD_550_, optical density at 550 nm). Biofilm formation activity was normalized by dividing OD_600_ of each borosilicate glass tube. * indicates that the biofilm forming unit of RR35 was significantly lower than that of PXO99A and cRR35 by Duncan’s test (*p* < 0.001).

**Figure 3 pathogens-09-00956-f003:**
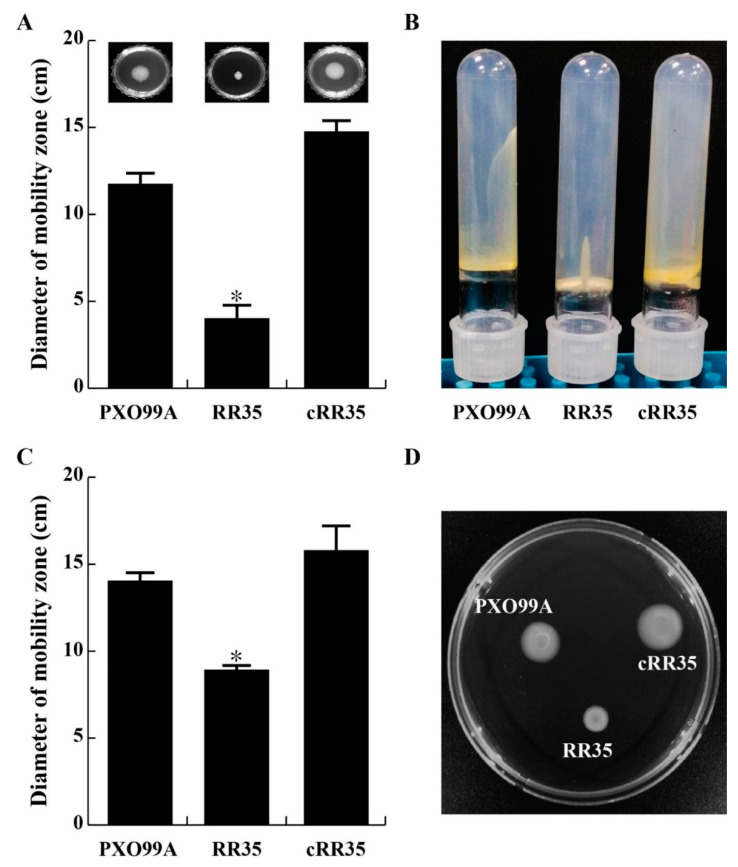
RR35 regulates bacterial motility of swimming, swarming, and twitching. (**A**) To test swimming motility, each strain was inoculated on 0.3% XOM2 agar at 28 °C for 7 days, and the diameters of migration zones were analyzed from three experiments. (**B**) Twitching motility was examined using borosilicate tubes containing 1% agar, followed by 7 days of incubation at 28 °C. (**C**,**D**) Swarming motility was examined at the same conditions as for swimming but with 0.5% XOM2 agar, where the RR35 mutant displayed altered swarming motility on plates * indicates that the diameter of RR35 was significantly smaller than that of PXO99A and cRR35 by Duncan’s test (*p* < 0.001).

**Figure 4 pathogens-09-00956-f004:**
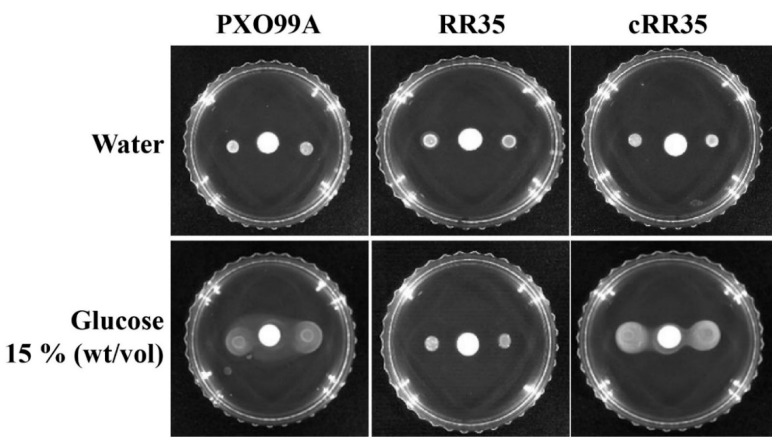
Effect of RR35 mutation on chemotaxis in *Xoo*. Chemotactic ability of the *Xoo* strains PXO99A, RR35, and cRR35 by calculating the difference between movements toward glucose. A Whatman paper disk saturated with glucose 15% (wt/vol) or water was placed at the center of the semisolid plate without a carbon source. Inoculated 2 µL sample from each strain was dropped from 1 cm above the paper disk, followed by 3 to 5 days of incubation at 28 °C. This experiment was repeated at least four times.

**Figure 5 pathogens-09-00956-f005:**
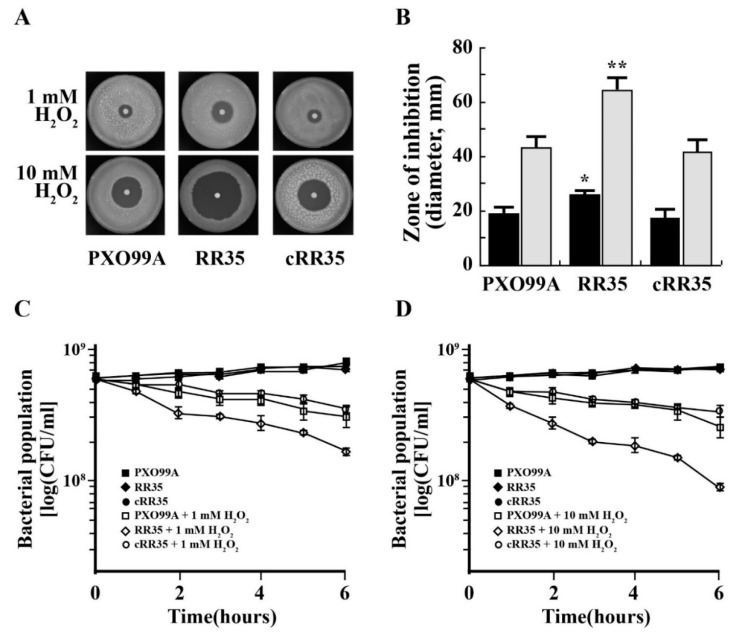
Hypersensitivity to hydrogen peroxide (H_2_O_2_). (**A**) The H_2_O_2_ sensibility was tested by disk diffusion assay. PXO99A, RR35, and cRR35 were cultured in XOM2 broth medium to OD_600_ = 1.0 and then mixed with soft PSA medium containing appropriate antibiotics for each strain. Paper disks loaded with 1 mM H_2_O_2_ or 10 mM H_2_O_2_ were placed in the middle of the plate, which was incubated at 28 °C for three days, and the diameters of inhibitory zone of strains were observed. (**B**) Inhibition zone diameters from the disk diffusion assays shown in (**A**). The experiment was performed as two/three independent biological replicates. Data with *p*-value < 0.001 are indicated with two asterisks, whereas data of *p*-value between 0.01 and 0.001 are indicated with an asterisk. Typical survival of PXO99A, RR35, and cRR35 grown in XOM2 broth media treated with 0 mM H_2_O_2_, 1 mM H_2_O_2_ (**C**), and 10 mM H_2_O_2_ (**D**) for 6 h. All experiments were repeated at least three times with consistent results.

**Figure 6 pathogens-09-00956-f006:**
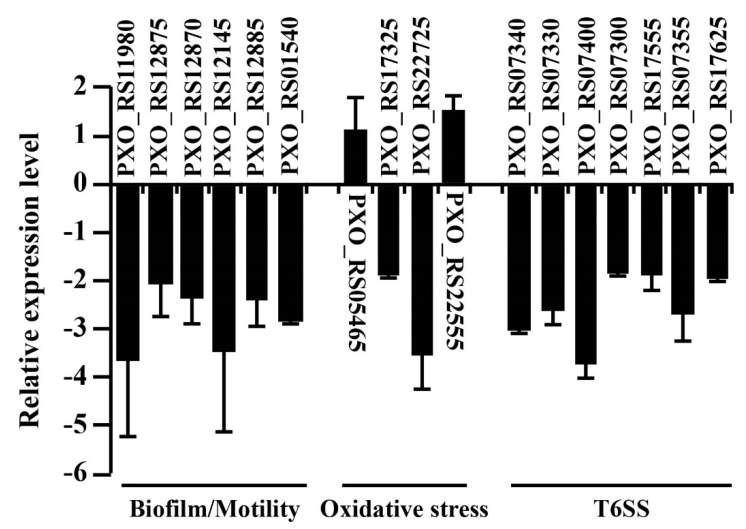
Relative expression of virulence factors of PXO99A and RR35. RNA samples were extracted from bacteria grown on XOM2 using RNeasy Plus Mini Kit (Qiagen, Valencia, CA, USA), and 40 ng of cDNA were used for quantitative real-time-PCR. 16S rRNA was used as a reference gene in each experiment. Bars are mean ± SD (n = 3). All experiments were repeated three times with high consistency, and results from one experiment are presented.

**Table 1 pathogens-09-00956-t001:** RNA sequencing analysis of transcripts up- or down-regulated in RR35.

Gene ID	Log2 Fold Change	Gene Product ^a^
**Up-Regulated Genes**		
PXO_RS06900	2.01	Prepilin-type N-terminal cleavage/methylation domain-containing protein
PXO_RS01045	2.01	Hypothetical protein
PXO_RS07555	2.02	Hypothetical protein
PXO_RS03845	2.05	Hypothetical protein
PXO_RS15405	2.05	Hypothetical protein
PXO_RS17195	2.11	IS5 family transposase ISXo1
PXO_RS19940	2.17	S46 family peptidase
PXO_RS21615	2.20	Adhesin
PXO_RS21300	2.21	Hypothetical protein
PXO_RS06905	2.22	Membrane protein
PXO_RS14325	2.24	Hypothetical protein
PXO_RS23085	2.28	Hypothetical protein
PXO_RS17240	2.29	DUF2946 domain-containing protein
PXO_RS25705	2.29	Hypothetical protein
PXO_RS23540	2.29	Hypothetical protein
PXO_RS23950	2.34	Hypothetical protein
PXO_RS20635	2.48	Methylisocitrate lyase
PXO_RS16940	2.50	DUF378 domain-containing protein
PXO_RS06660	2.59	MerC domain-containing protein
PXO_RS25610	2.62	Hypothetical protein
PXO_RS05075	2.80	Hemin uptake protein HemP
PXO_RS19360	4.08	C4-dicarboxylate transporter
PXO_RS04930	4.63	Hypothetical protein
PXO_RS17235	10.53	TonB-dependent receptor
PXO_RS17230	11.71	PepSY domain-containing protein
**Down-Regulated Genes**		
PXO_RS20535	−25.86	Response regulator
PXO_RS19530	−3.86	Arsenate reductase (glutaredoxin)
PXO_RS02330	−3.64	Hypothetical protein
PXO_RS27385	−3.54	DUF1905 domain-containing protein
PXO_RS05470	−3.48	alkyl hydroperoxide reductase subunit F
PXO_RS01535	−3.48	Acyl-CoA desaturase
PXO_RS03225	−3.36	Hypothetical protein
PXO_RS14005	−3.06	Hypothetical protein
PXO_RS01530	−2.99	Ferredoxin reductase
PXO_RS06115	−2.98	Arc family DNA-binding protein
PXO_RS13385	−2.68	Acyl-CoA dehydrogenase
PXO_RS24995	−2.68	IS5/IS1182 family transposase
PXO_RS00965	−2.67	Type VI secretion system tip protein VgrG
PXO_RS18385	−2.65	DNA transport competence protein
PXO_RS02070	−2.57	FMN reductase
PXO_RS26485	−2.53	Hypothetical protein
PXO_RS02075	−2.53	DUF1852 domain-containing protein
PXO_RS17835	−2.46	Holliday junction resolvase RuvX
PXO_RS16000	−2.45	Fimbrial protein
PXO_RS15525	−2.43	FAD-dependent monooxygenase
PXO_RS21890	−2.33	Hypothetical protein
PXO_RS13625	−2.29	Nucleotide exchange factor GrpE
PXO_RS17855	−2.27	DNA-3-methyladenine glycosylase I
PXO_RS23120	−2.26	3-Dehydroquinate dehydratase
PXO_RS09085	−2.25	Protease HtpX
PXO_RS13400	−2.24	5-Methyltetrahydrofolate--homocysteine methyltransferase
PXO_RS10260	−2.22	Hypothetical protein
PXO_RS27690	−2.19	IS5/IS1182 family transposase
PXO_RS23860	−2.18	IS630 family transposase
PXO_RS09410	−2.14	Cytochrome bd-I oxidase subunit CydX
PXO_RS17955	−2.14	Cysteine desulfurase
PXO_RS04055	−2.11	HslU--HslV peptidase proteolytic subunit
PXO_RS08660	−2.10	LacI family DNA-binding transcriptional regulator
PXO_RS02845	−2.10	Hypothetical protein
PXO_RS02305	−2.09	Molecular chaperone GroEL
PXO_RS17940	−2.08	Fe-S cluster assembly protein SufB
PXO_RS08655	−2.05	Phosphoenolpyruvate--protein phosphotransferase
PXO_RS17965	−2.05	Non-heme iron oxygenase ferredoxin subunit
PXO_RS11120	−2.01	DNA-binding response regulator
PXO_RS02770	−2.01	IS630 family transposase

^a^ DUF, domain of unknown Function; IS, insertion sequence element; PepSY domain, M4 peptidase inhibitory domain.

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
