# Peer review of "PXO_RS20535, Encoding a Novel Response Regulator, Is Required for Chemotactic Motility, Biofilm Formation, and Tolerance to Oxidative Stress in Xanthomonas oryzae pv. oryzae"

_pathogens, 2020, doi:10.3390/pathogens9110956_

Round 1

Reviewer 1 Report

This manuscript has provided critical information to comprehensively explain the integrated mechanism of pathogenesis regulated by TCSs in Xoo. In general, the experiments were well designed and conducted. I only have the following concerns.

  1. For all the q-RT-PCR or any other stress treatments, are you measuring “the difference” or “the difference in difference”? {The difference in difference (or "double difference") estimator is defined as the difference in average outcome in the treatment group before and after treatment minus the difference in average outcome in the control group before and after treatment3: it is literally a "difference of differences."}
  2. When you determine the DEGs, have you normalized them against the control so that we can exclude the possibility that the differentiated gene expression is caused by the circadian regulation?

Author Response

Reviewers’ Comments & Authors’ Replies

Manuscript ID.             pathogens-982481

Title                                 PXO_RS20535, encoding a novel response regulator, is required for chemotactic motility, biofilm formation, and tolerance to oxidative stress in Xanthomonas oryzae pv. oryzae

Authors                          Abdulwahab Antar, Mi-Ae Lee, Youngchul Yoo, Man-Ho Cho, and Sang-Won Lee

We sincerely appreciate the editors for their valuable comments on the previous version of our manuscript. Their review greatly helped us in improving its quality. In this response letter, we answer reviewers’ questions and summarize the edits we made according to reviewers’ suggestions. We use the “track changes” function of MS word to distinguish the updated sections. Once again, we thank all those who participated in the review for their time and effort on our manuscript.

Responses to Reviewer 1’s comments

This manuscript has provided critical information to comprehensively explain the integrated mechanism of pathogenesis regulated by TCSs in Xoo. In general, the experiments were well designed and conducted. I only have the following concerns.

  1. For all the q-RT-PCR or any other stress treatments, are you measuring “the difference” or “the difference in difference”? {The difference in difference (or "double difference") estimator is defined as the difference in average outcome in the treatment group before and after treatment minus the difference in average outcome in the control group before and after treatment3: it is literally a "difference of differences."}

All data for the bar graphs in our manuscript mean “the difference”, not “difference in differences (DID)”. All data from experiments in this manuscript is comparative between a wild type bacterial strain (PXO99A) and a knock-out mutant strain (RR35) that does not have a gene (rr35) in the PXO99A strain. Both strains (PXO99A and mutant) had no treatments in each experiment, thus a “difference in difference” for before and after treatment does not exist. Both strains were left in identical conditions and were observed the difference in specific phenotypes between two strains. Such an approach using a gene knock-out mutant strain is commonly used to study of the gene function. In this study, we characterized the gene (rr35) function in Xoo through investigation of changes in phenotype and gene expression (q-RT-PCR) of the gene knock-out mutant strain. We therefore measure the average outcome of the three sets of PXO99A strain and compare it to the average outcome of the three sets of RR35 mutant strain. All graphs simply show the average number from the three sets of PXO99A and the average number from the three sets of RR35 mutant in given conditions. In the case of q-RT-PCR, bars show expression level of each gene in the mutant strain relative to the ones in the PXO99A strain. These experiments

To elaborate on our experiment’s measurement methods, we provide an example. In the case of biofilm formation, we measured average biofilm amount from three set of each strain (PXO99A, RR35, and cRR35) in a given condition and this experiments repeated three times with consistent results. Figure 2 shows one of the results of three repeated experiments.

The measurement method (“the difference”) in our experiment is very common in studies that use mutant strains to investigate gene function.

We hope this answer will clear the reviewer's question.

  1. When you determine the DEGs, have you normalized them against the control so that we can exclude the possibility that the differentiated gene expression is caused by the circadian regulation?

The normalization method is RPKM (Reads Per Kilobase Million) as described in the previous manuscript. Through RNA-sequencing, it is possible to check the expression level of each gene or transcript of each sample with the number of mapped reads. However, to define the expression level by the number of mapped reads, the size of the sequencing data may be different for each sample, and the number of mapped reads varies according to the length of the gene or transcript. So a normalization process is very essential.

FYI, three metrics attempt are commonly used to normalize for sequencing depth and gene length in RNA sequencing analysis. Those are RPKM used in this study, FPKM (Fragments Per Kilobase Million), and TPM (Transcripts Per Kilobase Million). The method of RPKM is follow: 1. Count up the total reads in a sample and divide that number by 1,000,000 – this is our “per million” scaling factor. 2. Divide the read counts by the “per million” scaling factor. This normalizes for sequencing depth, giving you reads per million (RPM) 3. Divide the RPM values by the length of the gene, in kilobases. RNA-sequencing and the data analysis including data normalization in this study was carried out in Macrogen Inc. (Seoul, South Korea). If our study is focusing on developing the analyzing method of RNA sequencing data, it should be in. Our report focusing on functional characterization of a response regulator gene in a phyto-pathogenic bacteria. We so think that the detailed method for RPKM is not necessary in this manuscript.

Anyway, DEGs in this study were determined with comparative analysis using normalized data by RPKM. So we can exclude the differential expression of genes caused by different sequencing size in each sample and from gene (or transcript) length.

However the reviewers question is little confusing for us. “Can we exclude the possibility that the differentiated gene expression is caused by the circadian regulation?” If there is a circadian regulation in Xoo and if the RR35 is not involved in the circadian regulation, expression of the genes regulated by circadian rhythm should be no different between two strains. So the genes would be out of interest. If there is a circadian regulation in Xoo and the RR35 is involved in the circadian regulation, expression of the genes regulated by circadian rhythm should be different between PXO99A and RR35 mutant strain. It would be very interesting for us and the gene could be a target for study. However, as we know, there is not yet any reports that Xoo genes expression is regulated by circadian rhythm. We hope this answer will clear the reviewer's question.

Reviewer 2 Report

Line 15: „many changes in phenotypes of a knockout mutant“ ? it should be written more precisely, at least mention which are the most important ones.
Line 18: „was highly sensitive in present of H2O2“ – sensitive to what?

Lines 83-85: please, provide more details: pot size, greenhouse conditions, soil type etc.
Line 161: please, clarify how did you prepare the glucose disk.
Line 169: please, clarify the assay method. Which petri-dish have you used? 5 µl of H2O2 it seems to be too small volume, with probably, a high gradient in the disk.
Lines 206-209: please, provide some logical connection between part 3.1 and 3.2. Explain to the reader why you choose these strains for testing, how these strains were generated, etc.
Line 214: „reduced virulence observed in the RR35 mutant strain was a specific phenotype“ I am not sure about specific phenotype. Virulence is not a phenotype.
It will be great if you can write a short outlook for the future direction.

Author Response

Reviewers’ Comments & Authors’ Replies

Manuscript ID.             pathogens-982481

Title                                 PXO_RS20535, encoding a novel response regulator, is required for chemotactic motility, biofilm formation, and tolerance to oxidative stress in Xanthomonas oryzae pv. oryzae

Authors                          Abdulwahab Antar, Mi-Ae Lee, Youngchul Yoo, Man-Ho Cho, and Sang-Won Lee

We sincerely appreciate the editors for their valuable comments on the previous version of our manuscript. Their review greatly helped us in improving its quality. In this response letter, we answer reviewers’ questions and summarize the edits we made according to reviewers’ suggestions. We use the “track changes” function of MS word to distinguish the updated sections. Once again, we thank all those who participated in the review for their time and effort on our manuscript.

Responses to Reviewer 2’s comments

Line 15: “many changes in phenotypes of a knockout mutant“ ? it should be written more precisely, at least mention which are the most important ones.

As this reviewer suggested, the sentence was changed in revised manuscript (line 15 to 16) as below:

“In this study, we observed changes in virulence, biofilm formation, motility, chemotaxis, and tolerance against oxidative stress of a knockout mutant strain for the PXO_RS20535 gene, encoding an orphan response regulator (RR).

Line 18: „was highly sensitive in present of H2O2“ – sensitive to what?

As this reviewer pointed, the sentence was changed to clarify in revised manuscript (line 19 to 20) as below:

“Furthermore, the mutant strain lost glucose-guided movement and showed clear diminution of growth and survival in the presence of H2O2.”

Lines 83-85: please, provide more details: pot size, greenhouse conditions, soil type etc.

As this reviewer suggested, we provide detailed information about pot size, greenhouse conditions, soil type etc. in the section “2.1.. Biological Materials and Growth Conditions” in M & M of revised manuscript as below:

“Dong-jin (a japonica rice cultivar compatible to PXO99A) seeds were germinated in water at room temperature for 3 to 4 days, transferred into pots (12 cm in diameter and 11 cm in height) filled with soil supplied from rice paddy fields, and then grown in a greenhouse for five weeks (28 °C day /22 °C night).”

Line 161: please, clarify how did you prepare the glucose disk.

As this reviewer suggested, we detail the method of how we prepared the glucose disc in section “2.7.. Chemotaxis assay” in M & M of revised manuscript (line 163 to 167) as below:

“Next, sterilized filter discs (5 mm diameter) were infiltrated with 15% (wt/vol) glucose or water under vacuum and placed in the center of a soft agar plate (35 x 10mm, SPL life science, South Korea). 2 μL of each cell strain was dropped on either side, 1 cm away from the paper disc. The plates were placed at 28 °C, and movement of bacteria toward the disc was monitored for 3 to 5 days.”

Line 169: please, clarify the assay method. Which petri-dish have you used? 5 µl of H2O2 it seems to be too small volume, with probably, a high gradient in the disk.

As this reviewer suggests, we clarified the assay method for H2O2 tolerance assay in section 2. 8. in M & M as below:

First, size and company of the used petri-dish are provided in line 175 of revised manuscript.

5 µl of H2O2 seems to be a small volume as this reviewer suggests, but it was enough to show the growth inhibition on the plates in each of the repeated experiments.

Yes, there may be a high gradient in the disk from the drop of 5 µl of H2O2. The concentration will gradually decrease away from the dropped point, but we do not know the exact concentration of hydrogen peroxide at each point away from the dropped point. Additionally, the gradient will be similar or even the same in each plate because all plate conditions, such as composition of the medium, used volume (20 mL) added to each petri-dish, and making date were identical in each set of experiments. Even if generated gradients were to vary slightly for plates used to test PXO99A and the mutant strains, repeated experiments with independently prepared strains and plates had very consistent results: RR35 strain always made a bigger halo zone than PXO99A and cRR35. This differences in halo size between the strains are the main point of this experiments, and is clearly depicted in Figure 5.A.

Lines 206-209: please, provide some logical connection between part 3.1 and 3.2. Explain to the reader why you choose these strains for testing, how these strains were generated, etc.

As this reviewer suggested, we switched part 3.1 and 3.2 of the previous manuscript in this revised version and added a sentence at the start of the updated 3.1 as below:

“From our mutant screening, we selected six rr genes, associated with Xoo virulence [23]. The rr35 gene used in this study is one of the six. Consistent…”

Line 214: „reduced virulence observed in the RR35 mutant strain was a specific phenotype“ I am not sure about specific phenotype. Virulence is not a phenotype.

The experiment was to verify that the mutant strain has no effect by polar mutation. So we described this lack of effect as “a specific phenotype” in the previous manuscript. However, as the reviewer pointed out, our way of expression may not be clear.

In the revised version, the sentence was changed to: “This result indicates that the cRR35 strain restored virulence, and the reduced virulence observed in the RR35 mutant strain was due to knockout of the PXO_RS20535 gene, not to unexpected mutation of adjacent genes during homologous recombination in mutagenesis.”

It will be great if you can write a short outlook for the future direction.

Short future plan was indeed described at the end of “Discussion”, line 419 to 495 of the previous version. We added a little more detail and changed the writing to clearly show our future direction. Below is the update included in the new version of our manuscript.

“As mentioned in our previous reports [19, 23], we are elucidating the pathogenic mechanisms of the six RRs that we recently discovered. PXO_RS20535, in this study, is the second-characterized RR gene after detR [19] and research on the remaining four will be shortly completed. Furthermore, to accumulate data on the interrelationships between all RRs in gene expression, we are carrying out large-scale transcriptome analysis, such as RNA-sequencing using RR mutant strains. These further studies will undoubtedly contribute to uncovering the integrated mechanism of TCSs for Xoo pathogenesis.”